# Concomitant Infection of *Helicobacter pylori* and Intestinal Parasites in Adults Attending a Referral Centre for Parasitic Infections in North Eastern Italy

**DOI:** 10.3390/jcm9082366

**Published:** 2020-07-24

**Authors:** Elena Pomari, Tamara Ursini, Ronaldo Silva, Martina Leonardi, Marco Ligozzi, Andrea Angheben

**Affiliations:** 1Department of Infectious-Tropical Diseases and Microbiology, IRCCS Sacro Cuore Don Calabria Hospital, Negrar di Valpolicella, 37024 Verona, Italy; tamara.ursini@sacrocuore.it (T.U.); ronaldo.silva@sacrocuore.it (R.S.); martina.leonardi@sacrocuore.it (M.L.); andrea.angheben@sacrocuore.it (A.A.); 2Department of Diagnostics and Public Health, University of Verona, 37134 Verona, Italy; marco.ligozzi@univr.it

**Keywords:** *Helicobacter pylori*, intestinal parasites, coinfection, Italy

## Abstract

Background: *Helicobacter pylori* and intestinal parasites are estimated to infect with high burden worldwide. However, their concomitant infections are poorly determined in industrialized countries, such as Italy. In this study we aim at describing the presence of *H. pylori* as well as the proportion of coinfections with intestinal parasites among subjects who attended a referral center for tropical diseases in Northern Italy. Methods: This was a case-control study. Screening for *H. pylori* and parasites was performed on stool samples of 93 adults from different geographical origin (Africa, Asia, South-America, East-Europe and Italy). *H. pylori* infection was examined by CLIA and its *cagA* positivity was determined by rtPCR. Intestinal parasites (i.e., protozoa and helminths) were examined by microscopy and rtPCR. Results: Sixty-one out of 93 patients (66%) were positive to *H. pylori* and 31 (33%) were *cagA*+. Among *H. pylori* positives, 45 (74%) had a concomitant infection. The coinfection *H. pylori*–*Blastocystis* was the most frequent one, followed by *H. pylori–E. coli*. Multivariable logistic regression showed that positivity to *H. pylori* was associated with having a coinfection. Conclusion: Our data suggested that *H. pylori* and intestinal parasitic infections are fairly common in subjects who attended a referral center for tropical diseases in Northern Italy. The high rate of *H. pylori* infection, and especially the positivity to the virulent *cagA*+, should be taken into consideration in subjects undergoing screening for parasitic infections.

## 1. Introduction

*Helicobacter pylori* and intestinal parasites are infectious agents of worldwide public health importance and common causes of gastrointestinal (GI) discomforts [1,2,3,4,5,6,7]. *H. pylori* is a Gram-negative bacterium identified as a major cause of peptic ulcer and gastric cancer [8]. It is estimated that *H. pylori* infects more than 50% of the world population [9,10] with the highest burden among individuals living in low and middle-income countries (LMICs) [9,10,11,12,13,14]. Similarly, intestinal parasites affect millions of people globally [15,16,17], leading to a high probability of coinfection with *H. pylori*. Over the past decade, an impressive increase in migration flow has occurred in European countries including Italy, but little data about the rate of concomitant *H. pylori*-intestinal parasites infections in travelers is available [18,19]. In Italy, *H. pylori* and concomitant infections are not routinely screened in travelers, with consequent possible diagnostic pitfalls in case of symptoms and lack of the local transmission potential [20]. In particular, previous studies reported high prevalence of these coinfections among people of LMICs with poor healthcare and hygiene [21,22,23,24], but further investigations about the rate of coinfection *H. pylori*-intestinal parasites and the potential *H. pylori* pathogenic factors may support the improvement of screening and diagnosis criteria in industrialized countries [18,20,25,26,27,28,29,30]. Thus, this study aims at improving the knowledge regarding the rate of *H. pylori* and intestinal parasites (protozoa and helminths) by describing these concomitant infections among adult subjects who were screened for intestinal parasitic infections at our reference center for tropical diseases in North Eastern Italy. For the purpose of our investigation, the subjects included in this study were screened also for *H. pylori* and for its *cagA* virulence factor. Furthermore, the association between concomitant infections and the presence of GI symptoms was evaluated. 

## 2. Materials and Methods

### 2.1. Setting and Participants

This case-control study was performed at the Department of Infectious-Tropical Diseases and Microbiology (DITM), IRCCS Sacro Cuore Don Calabria, Negrar di Valpolicella, a referral center for tropical medicine in Veneto region. Data of subjects attending our center for the screening for intestinal parasites and *H. pylori* from March 2018 to May 2019 were assessed for eligibility. 

Eligibility criteria. Adult subjects who were tested for intestinal parasites and *H. pylori* were included. Subjects who were not on antibiotic and anti-parasitic drugs within four weeks prior to the screening.

### 2.2. Ethics Approval and Consent to Participate

This study (No. 39169/2019) was approved by the competent Ethics Committee for Clinical Research of Verona and Rovigo Provinces. The study was conducted in accordance with the Declaration of Helsinki and all the subjects signed up an informed consent.

### 2.3. Data Collection

Main outcomes were stool antigen test (SAT) results and PCR_*cagA* for *H. pylori* infection; and microscopy and PCR results for intestinal parasites. Clinical and demographic data were also collected. The detailed dataset is reported in the Appendix A.

### 2.4. H. pylori Detection by Stool Antigen Test

All stool samples were screened for *H. pylori* infection using a chemiluminescent immunoassay (CLIA) intended for the qualitative determination of *H. pylori* stool antigen in human feces (LIAISON® *H. pylori* SA, DiaSorin, Saluggia, Italy) which is approved nationally and used for non-invasive detection of *H. pylori* infection, following the manufacturer’s instruction.

### 2.5. H. pylori Caga Virulence Factor Analysis by Real-Time PCR

The total DNA was extracted from 200 mg of stool using MagNA Pure LC 2.0 Instrument (Roche, Monza, Italy). In each sample, Phocine Herpes Virus type-1 (PhHV-1) was added as internal control for the isolation and amplification steps, as described previously [31,32]. All the amplification reactions were performed using 5 µL of DNA and using SsoAdvanced universal probes supermix (BioRad, Milan, Italy), primers cagA-F 5′-TCAAGAACCAGTTCCCCATGTC-3′ and cagA-R 5′-TCTCTAGCTTCAGGCGGTAAGC-3′, and probe HEX-5′-ACCAGATATAGCCACTACC-3′. 

The program consisted of an initial step of 3 min at 95 °C followed by 50 cycles of 15 sec at 95 °C, 30 sec at 58 °C and 30 sec at 72 °C. All reactions and data analyses were performed on CFX96 system (BioRad, Milan, Italy).

### 2.6. Intestinal Parasites Detection by Real-Time PCR

According to the routine procedure of our laboratory, molecular diagnostic screening for intestinal parasites was performed by four separate multiplex rt-PCRs for *Entamoeba histolytica*—*Entamoeba dispar*—*Cryptosporidium* spp., for *Giardia intestinalis*—*Dientamoeba fragilis*—*Blastocystis* spp., for *Strongyloides stercoralis*—*Schistosoma* spp—*Hymenolepis nana* and for *Necator americanus*—*Ascaris lumbricoides*—*Ancylostoma duodenale*—*Trichuris trichiura*. Multiplex rt-PCRs were performed adapting the reported protocols [33,34,35,36,37,38,39,40,41]. The total DNA was extracted from 200 mg of stool using MagNA Pure LC 2.0 Instrument (Roche, Monza, Italy). In each sample, PhHV-1 was added as internal control for the isolation and amplification steps, as described previously [31,32]. All the amplification reactions were performed using 5 µL of DNA and using SsoAdvanced universal probes supermix (BioRad, Milan, Italy). The primers/probe sets are reported in Appendix A. The program consisted of an initial step of 3 min at 95 °C followed by 40 cycles of 15 sec at 95 °C, 30 sec at 60 °C and 30 sec at 72 °C. All reactions and data analyses were performed on CFX96 system (BioRad, Milan, Italy). 

### 2.7. Intestinal Parasites Detection by Microscopy

Microscopy observation was performed on stool for a wide investigation of parasites including those screened by real-time PCR (as described above) and also *Entamoeba coli*, *Entamoeba hartmanni*, *Entamoeba nana*, *Iodamoeba butschlii*. Stool samples were microscopically examined for ova and cysts in wet mount preparations of a formal-ether concentrate. Coprocultures were used for the specific diagnosis of *S. stercoralis* and hookworms.

### 2.8. Data Analysis

Data analysis was performed using SAS software version 9.4. Collected data were summarized using descriptive statistics. Age was reported as median (Mdn) and first and third quartile (IQR). PCR and microscopy results were combined using OR rule when they targeted the same intestinal parasite. Laboratory results were presented in contingency table of absolute and relative frequencies. We used Chi-square and Fisher tests to assess significant association between presence of infections and demographic, clinical variables. The statistical significance level was fixed at 5% and estimated parameters were reported with 95% confidence intervals.

## 3. Results

### 3.1. Baseline Characteristics

Data of 93 subjects were analyzed. For the details see the study flow diagram in Figure 1 and Table 1 (see also Appendix A). Seventy-one (76%) were male (Mdn age = 27, IQR = 21–38) and 22 were female (Mdn age = 40, IQR = 32–45). Eighty-four percent of subjects were from the African continent. At the time of the screening 32/93 (34%) reported GI symptoms, of which 19/32 (59%) of subjects had upper abdominal pain, 10/32 (31%) epigastric pain, 7/32 (14%) diarrhea, and 4/32 (13%) gastritis/duodenitis. Sixty-one (66%) subjects were infected with *H. pylori* of which 31/61 (51%) were *cagA*+; 29 of these 31 subjects were from Africa. The infection rate was higher in males than in females (*X^2^*(1, *N* = 93) = 3.95, *p* = 0.0070). The most significant symptom in the *H. pylori* positives was the epigastric pain. Four subjects underwent endoscopy (see Table 1 for details) and no significant difference was observed in the findings compared to the *H. pylori* negatives. Among the *H. pylori* positives, no significant differences were observed between the *cagA*+ and *cagA*- groups. See Table 2 and Appendix A, and Appendix A for details. 

### 3.2. Concomitant Infections

Forty-five out of 61 *H. pylori* positives (74%) had a concomitant infection (Table 3). The association between *H. pylori* and concomitant parasitic infection was significant in the entire cohort (*X^2^* (1, *N* = 93) = 6.63, *p* = 0.01) with odds of having co-infection with other parasites of 3.2 (95% CI = 1.2977–7.8292) compared to that of the *H. pylori* negative group. Among the *H. pylori* positives the concomitant infection with *Blastocystis* was the most frequent with 41/61 (67%) of subjects infected, followed by *E. coli* with 12/61 (20%) subjects. The odds of having *Blastocystis* infection was significantly higher (odds ratio, 3.4, 95% confidence interval, 1.39–8.35) in subjects with *H. pylori*. No significant difference in having or not concomitant parasitic infections was observed when the comparison was made by gender and age. See Appendix A for subgroup analyses. 

### 3.3. Clinical Features and Infections

Concerning the association between *H. pylori* infection and GI symptoms, we found

*X*^2^ (1, N = 93) = 5.09 (*p* = 0.0240) with odds of having symptoms of 0.36 (95% CI = 0.145–0.873). For example, in the most frequent concomitant pair of infections *H. pylori*-*Blastocystis*, 31/41 (76%) of subjects were asymptomatic. Regarding all other parasites, their prevalence was too low to justify any exploratory analysis on associated symptoms. In Table 4 and Table 5 are reported the frequency of GI symptoms by type of infection in the whole population and in the subset of subjects positive to *H. pylori*, respectively. No significant difference in having or not concomitant parasitic infections was observed when the comparison was made by gender and age in *H. pylori* positives as well as in *H. pylori* negatives. Among the *H. pylori* positives, the endoscopy findings were significantly different between having or not concomitant infection. However, the number of subjects undergoing digestive endoscopy were limited. See Appendix A for subgroup analyses. 

## 4. Discussion

The present study described the occurrence of *H. pylori* infection along with intestinal parasites concomitant infections in a cohort of adults attending a tropical diseases center in North Eastern Italy. The majority of subjects came from Africa, mirroring the migration pattern to this area of Italy. The majority of individuals were infected with *H. pylori* and most of them were *cagA+*, suggesting potential mechanisms of aiding the entry of other pathogens [42]. Our data are consistent with the results of a systematic review by Morais and colleagues [43] as well as the findings of a recent Dutch study, highlighting that *H. pylori* seroprevalence among first-generation migrants was high and remains elevated among second-generation migrants (i.e., those born in the Netherlands) [44]. Moreover, in our cohort, a higher rate of *H. pylori* infection was observed among intestinal parasites infected individuals compared to uninfected subjects. The most frequent intestinal parasites were protozoa, and specifically the association of *H. pylori* with *Blastocystis* spp. was the most prevalent. Even though the association was not statistically significant, a higher proportion of subjects infected with *E. coli* and *E. nana* were found among *H. pylori*-infected participants. A study conducted in 115 irritable bowel syndrome (IBS) patients from Egypt showed that 27% of *H. pylori*-infected individuals were coinfected with *Blastocystis* [45]. A marked association of *Blastocystis* and *H. pylori* (67% coinfection) was also recently found in Pakistani subjects with chronic diarrhea [46]. A previous study conducted at our center aimed at evaluating *Blastocystis* prevalence and subtypes revealed that 46% of *Blastocystis*-positive samples had coinfection, with *D. fragilis* being the most frequent co-infecting parasite [47]. *G. intestinalis*, *E. hystolytica/dispar*, and *H. pylori* are considered the most common infectious pathogens affecting humans in LMIC [48]. A similar previous study that evaluated 363 adult patients from Ethiopia showed that *G. lamblia* prevalence (22.3%) was significantly higher among *H. pylori*-infected participants [12]. Similarly, in 427 non-symptomatic children from Uganda with 44.3% prevalence of *H. pylori, G. intestinalis* was the dominant (20.1%) concomitant parasite [49]. Moreira et al. did not find significant association between *E. histolytica* infection and *H. pylori* seropositivity [48]. However, Torres et al. showed a significantly lower prevalence of *H. pylori* infection among adults carrying *E. histolytica* compared to those who were negative to the parasite [50]. Hence, this needs well designed cohort type research to ascertain the presence of an association, if any. In our study, both *Blastocystis* and *H. pylori* were commonly found in the asymptomatic subjects, suggesting that the detection of the microbes in stool from subjects with GI symptoms was likely a chance of finding and not a causative role in GI pathophysiology. Similarly, a recent population case study conducted in Denmark showed that *Blastocystis* and *D. fragilis* were detected in a greater proportion of fecal samples from asymptomatic population than from subjects with IBS symptoms [51]. 

The use of SAT testing for our cohort vs. the majority of the above discussed studies using serology, permitted more accurate *H. pylori* measurement as it detects active infection. The advantage of SAT over other diagnostic tests, is that it is a rapid, easily accessible, and more acceptable as non-invasive. However, a potential limitation of our study regarding the diagnosis for *H. pylori* only on stool needs to be acknowledged, indeed further prospective studies including additional histological findings on gastric biopsy would be more informative. Our study has other limitations mainly because of the retrospective design. Primarily, no formal sample size calculation was performed. Some variables were not standardized and the endoscopy findings were available in a limited number of subjects. Moreover, concerning the proportion of who was included, it may not be representative of migrant population in other Italian (or European) centers. Second, no migration history was recorded and no all participants were newly arrived. Thus, in order to conclude important clinical implications among travelers (autochthonous and not) and Italian non travelers, further evaluations are warranted in a larger cohort of subjects. 

## 5. Conclusions

Our study showed a high rate of *H. pylori* infection as well as the rate of coinfection with intestinal parasites among a cohort of subjects screened at a referral center for tropical diseases. The presence of concomitant infections was fairly common in both GI symptomatic and asymptomatic subjects. High exposure to *H. pylori*, and especially to the more virulent *cagA+* strain, highlights the need for tailored preventive strategies among these people. Further studies are warranted to investigate *H. pylori* virulence factors that would support the clinical management of these subjects. Moreover, additional investigations might be important to better understand the mechanisms behind a possible association of the *H. pylori cagA*+ with other co-infections.

## Figures and Tables

**Figure 1 jcm-09-02366-f001:**
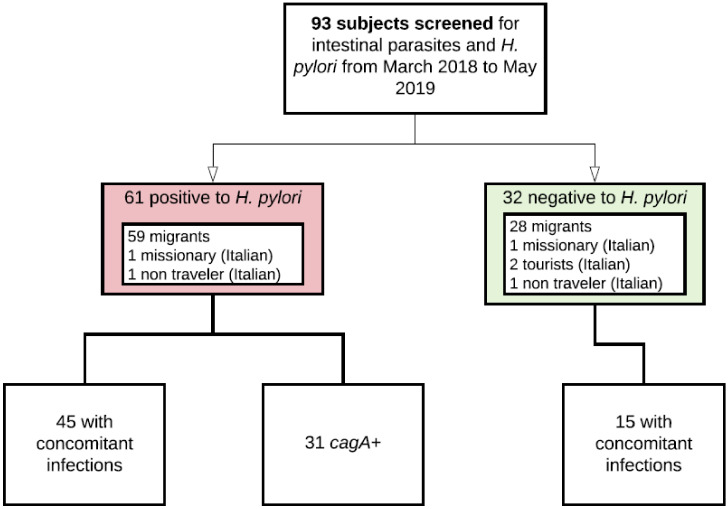
Study flow diagram.

**Table 1 jcm-09-02366-t001:** Baseline characteristics of the screened subjects (N = 93).

Variable		N (%) or Median (IQR)
**Age (Years)**		31 (23–41)
Sex	Female Male	22 (23.66) 71 (76.34)
Geo Origin	Africa	78 (83.87)
	Italy	6 (6.45)
	Asia	4 (4.30)
	South-America	3 (3.23)
	East-Europe	2 (2.15)
Clinical features	Abdominal pain Epigastric pain Diarrhea	19 (20.43) 10 (10.75) 7 (7.53)
Endoscopy findings	Chronic gastritis	1 (1.08)
	Chronic gastritis and erosive duodenitis	1 (1.08)
	Antral gastritis and bulbar duodenitis	2 (2.15)

**Table 2 jcm-09-02366-t002:** Baseline characteristics of the screened subjects (N = 93) between the two groups *H. pylori* positive and negatives. Categorical and continuous variables are presented as numbers (%) and medians (interquartile range), respectively.

Variable		*H. pylori* Positives (*N* = 61)	*H. pylori* Negatives (*N* = 32)	*p* Value
**Age (Years)**		27 (21–37)	39 (27.50–51)	0.0013
Sex				0.0050
	Female Male	9 (40.91) 52 (73.24)	13 (59.09) 19 (26.76)	
Geo Origin	Africa	53 (67.95)	25 (32.05)	-
	Italy	2 (33.33)	4 (66.67)	-
	Asia	2 (50)	2 (50)	-
	South-America	2 (66.67)	1 (33.33)	-
	East-Europe	2 (100)	-	-
Clinical features	Abdominal pain	7 (36.84)	12 (63.19)	0.0141
	Epigastric pain	8 (80)	2 (20)	0.0034
	Diarrhea	3 (42.86)	4 (57.14)	0.1808
Endoscopy findings				0.1366
	Chronic gastritis	1 (100)	-	
	Chronic gastritis and erosive duodenitis	-	1 (100)	
	Antral gastritis and bulbar duodenitis	-	2 (100)	

**Table 3 jcm-09-02366-t003:** Rate of intestinal parasitic infections and mixed infections found in the cohort.

Parasite	Positive Subjects *N* (%)	Parasite Rate with Concomitant *H. pylori* *n* (%)	Multi-Parasitic Infections with Concomitant *H. pylori* *n* (%)
*Blastocystis* spp.	53(56.99)	41 (77.36)	24 (58.54)
*E. coli*	12 (12.90)	12 (100)	12 (100)
*E. nana*	11 (11.83)	10 (90.91)	10 (100)
*S. mansoni*	9 (9.68)	9 (100)	9 (100)
*E. dispar*	6 (6.45)	6 (100)	6 (100)
*D. fragilis*	6 (6.45)	5 (83.33)	5 (100)
*E. histolytica*	4 (4.30)	4 (100)	4 (100)
*G. intestinalis*	4 (4.30)	2 (50.0)	2 (100)
*S. stercoralis*	4 (4.30)	3 (75.00)	3 (100)
*E. hartmanni*	3 (3.23)	3 (100)	3 (100)
*A. duodenale*	3 (3.23)	3 (100)	3 (100)
*I. butschlii*	2 (2.15)	2 (100)	2 (100)
*H. nana*	1 (1.08)	1 (100)	1 (100)

**Table 4 jcm-09-02366-t004:** Frequency of gastrointestinal (GI) symptoms by type of infection. *H. pylori* infection was detected by stool antigen test (SAT). Intestinal parasites were examined by real-time PCR and microscopy.

Infection	Total *N*	GI *n* (%)	Odds Ratio (95% CI)	*p*-Value
*H. pylori*	61	16 (26.23)	0.356 (0.145,0.873)	0.0240
*Blastocystis* spp.	53	16 (30.19)	0.649 (0.274,1.537)	0.3253
*E. coli*	12	2 (16.67)	0.340 (0070,1.657)	0.1819
*E. nana*	11	2 (18.18)	0.385 (0.078,1.902)	0.2416
*S. mansoni*	9	2 (22.22)	0.514 (0.100,2.634)	0.4250
*E. dispar*	6	1 (16.67)	0.361 (0.040,3.233)	0.3626
*D. fragilis*	6	3 (50.00)	2 (0.380,10.532)	0.4135
*N. americanus*	5	1 (20.00)	0.460 (0.049,4.294)	0.4954
*G. intestinalis*	4	2 (50.00)	1.967 (0.264,14.658)	0.5093
*S. stercoralis*	4	1 (25.00)	0.624 (0.062,6.250)	0.6880
*E. hartmanni*	3	1 (33.33)	0.952 (0.083,10.913)	0.9682
*I. butschlii*	2	1 (50.00)	1.935 (0.117,32.004)	0.6446

**Table 5 jcm-09-02366-t005:** Frequency of GI symptoms by type of infection in the subset of subjects positive to *H. pylori. H. pylori* infection was detected by SAT. Intestinal parasites were examined by real-time PCR and microscopy.

Mono Coinfetion with *H. pylori*	Total *n*	GI *n* (%)	Odds Ratio (95% CI)	*p* Value
*Blastocystis* spp.	41	10 (24.39)	0.753 (0.228,2.481)	0.6406
*E. coli*	12	2 (16.67)	0.500 (0.097,2.577)	0.4074
*E. nana*	10	1 (10.00)	0.267 (0.031,2.294)	0.2287
*S. mansoni*	9	2 (22.22)	0.776 (0.144,4.189)	0.7677
*E. dispar*	6	1 (16.67)	0.533 (0.057, 4.948)	0.5802
*D. fragilis*	5	3 (60.00)	4.962 (0.747,32.964)	0.0974
*E. hartmanni*	3	1 (33.33)	1.433 (0.121,16.969)	0.7753
*I. butschlii*	2	1(50.00)	2.933 (0.173,49.860)	0.4566
*G. intestinalis*	2	1 (50.00)	2.933 (0.173,49.860)	0.4566

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
