# Peer review of "Concomitant Infection of Helicobacter pylori and Intestinal Parasites in Adults Attending a Referral Centre for Parasitic Infections in North Eastern Italy"

_jcm, 2020, doi:10.3390/jcm9082366_

Round 1

Reviewer 1 Report

While this is an interesting study, and for the most part it was well done, I can see no new information contained within the manuscript.  The finding of significant association between H. pylori infection and intestinal parasitic infestation is interesting, but not really news.  I have a technical problem with the description of the rtPCR used in the study (and I am not a clinical microbiologist, but extensively familiar and experienced with rtPCR and other forms of PCR).  The methods claim that 5 microliters (a volume, not a mass) of total DNA extracted from fecal matter was used as template DNA in the rtPCR.  Unless the DNA was quantified and diluted such that each DNA sample possessed the identical quantity of DNA, a volumetric measure of input DNA is inappropriate as there may be varying levels of DNA in each sample.  Thus comparing results from different samples is problematic.  However, as the results reported are "plus or minus" rather than levels of infection (as it would seem that rtPCR would generate), I think the results are likely valid.

Reviewer 2 Report

The manuscript showcases the occurrence of intestinal co-infections concomitant with H.pylori: and their prevalence especially in migrants since certain parameters are not included in other countries' clinical evaluation. The authors advocate the inclusion of certain parameter which will go a long way to aid early detection. The flow of information looks adept, certain minor modifications for word choice errors, scientific jargon usage and grammatical need to be done for flawless readability.

  1. Line # 12- Please replace high prevalence. with an apt word
  2. Line #13- Please change rate with an apt alternative.
  3. Lines #32-35: Please rephrase for grammatical fitness.
  4. Line #48: Please rephrase for grammatical fitness.
  5. The study design does not quite appeal to its nomenclature of "retrospective study", please use an appropriate term.
  6. Lines 102-108: There is a change in verb tense in the whole para 2nd line onwards please rectify.
  7. Line #138: please change the word rate with prevalence or dominance or other word of your choice.
  8. Line #141- grammar change needed.
  9. Entamoeba coli has been generally reported to be non pathogenic however it does serve as a model organism indicating the prospective invasion of other pathogens.
  10. Line#161: It would help to propose a plausible explanation suggesting the cause of this difference.
  11. While reviewing the supplementary material, it struck that in the case of chronic H.Pylori infection with cagA. It will be helpful if the explanation for the persistence of these co-infections could be dealt with respect to the disease mechanism adopted by the co-infecting pathogens. H. pylori CagA causes significant changes to the membrane dynamics threatening the balance and possibly aiding the entry of other pathogens. 
